# Cross-modal Observation Hypothesis Inference

## ABSTRACT

Hypothesis inference, a sophisticated cognitive process that allows humans to construct plausible explanations for incomplete observations, is paramount to our ability to make sense of the world around us. Despite the universality of this skill, it remains under-explored within the context of multi-modal AI, which necessitates analyzing observation, recalling information in the mind, and generating explanations. In this work, we propose the **C**ross-modal **O**bservation hypothes**I**s i**N**ference task (**COIN**). Given a textual description of a partially observed event, COIN strives to recall the most probable event from the visual mind (video pool), and infer the subsequent action flow connecting the visual mind event and the observed textural event. To advance the development of this field, we propose a large-scale text-video dataset, Tex-COIN, that contains $39,796$ meticulously annotated hypothesis inference examples and auxiliary commonsense knowledge (appearance, clothing, action, etc.) for key video characters. Based on the proposed Tex-COIN dataset, we design a strong baseline, COINNet, which features two perspectives: 1) aligning temporally displaced textual observations with target videos via transformer-based multi-task learning, and 2) inferring the action flow with non-parametric graph-based inference grounded in graph theory. Extensive experiments on the Tex-COIN dataset validate the effectiveness of our COINNet by significantly outperforming the state-of-the-arts. The code is available [1], and the dataset will be released for further exploration.

## CCS CONCEPTS

• **Computing methodologies** → **Natural language processing**; **Computer vision**.

## KEYWORDS

Multi-modal Understanding, Hypothesis Inference

## 1 INTRODUCTION

Hypothesis inference aims at proposing the most probable hypotheses to elucidate incomplete observation, which encapsulates a central pillar of human cognitive abilities to understand our surroundings [43]. This form of reasoning presents a dynamic interplay of memory and inference, where the incomplete event observation triggers the recall of historical visual events in memory, fostering the gradual inference of a holistic hypothesis serving as the explanation. For

---

[1] https://anonymous.4open.science/r/COIN-F621/

example, when we observe *O: A burned boy is being treated in the hospital.* and visually recall that he previously cooked in an oven, we can gradually infer a complete event chain as the explanation *E: the boy cooked in an oven → he was burned → he put down the food → he cooled the wound → he hurriedly ran to the hospital.* Hypothesis inference, with its wide-ranging applications in human existence, enables us to perform better than machines in high-level reasoning and would be the most precious capacity for modern AI (*e.g.*, Judicial AI, Security AI). Yet, to the best of our knowledge, its integration into the realm of multi-modal AI systems — systems that strive to understand incomplete textual observations, visually recall related events, and infer the comprehensive chain of events — remains a frontier largely unexplored.

To facilitate the development in this AI field, we propose a novel task, **C**ross-modal **O**bservation hypothes**I**s i**N**ference (**COIN**). COIN draws its inspiration from the burgeoning advancements in the field of causal reasoning within the NLP community [3, 14] and further extends its reach into the multi-modal domain. The essence of COIN lies in its ability to recall the inception of an event from memory and infer a cohesive event flow that furnishes a robust explanation for an incomplete observation. Specifically, considering that 83% of the information memorized by the human brain is in the continuous visual form [46], we integrate a video pool to simulate human visual memory. The AI system we propose is tasked with not only retrieving the target video that serves as the event chain's starting point but also inferring the subsequent action flow (subsequent event inference) that emerges after this retrieved video event and before the observed textural event. Examining and adopting ideas for a relevant, well-established, yet distinct task, *i.e.* text-video retrieval, provides a robust foundation for the development of this new field. The COIN task introduces two key distinctive elements that lend it a unique complexity: **(1) Heterogeneous Alignment for Inference.** Information alignment is the foundation of solving the retrieval tasks such as our COIN task and the text-video retrieval task. However, in the context of the COIN task, the textual observations and the target video do not align temporally, nor do they correspond to the same event. Instead, they describe logically correlated events, creating a challenging yet intriguing landscape for cross-modal information alignment. The task complexity is further amplified when we consider the need for extracting observation-related information (*e.g.*, actions) in the video. **(2) Action Flow Inference.** There are rigorous logical connections between actions in the action flow. An error in the inference of any single action could potentially disrupt the correctness of subsequent actions. Action flow inference necessitates high fidelity in uncovering the relationships between actions and accurately predicting each one.

To promote the development of this field, we propose a new large-scale dataset, **Tex-COIN**, towards the two distinctive characteristics of the COIN task. The dataset consists of about $10,000$ carefully collected videos, with $39,796$ hypothesis inference samples meticulously annotated by annotators with strong logical abilities. Our Tex-COIN dataset contains the following targeted designs: **(1)**

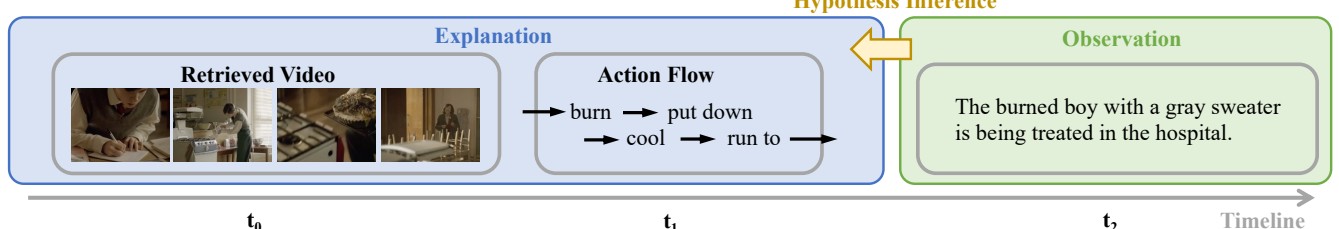

**Figure 1: An example of our COIN task, which aims to explain the observation by retrieving the target video (containing a video clip in which the target character cooks in an oven) and inferring the subsequent action flow.**

**Commonsense Knowledge Annotation.** We have incorporated a diverse range of commonsense knowledge annotations pertaining to the video characters described by the textural observation. These annotations, covering aspects such as appearance, attire, actions, and emotional state, are aimed at enhancing the model's heterogeneous alignment of retrieved video and textural observation. **(2) Action Flow Annotation.** For each video, annotators with notable logical capabilities are entrusted with annotating the incomplete textural observation, which envisions a future event, and the sequence of actions that transpire post the video event and preceding the textual event. In addition, the Tex-COIN dataset has the potential to advance broader task evaluations, such as temporally displaced text-to-video retrieval, thereby fostering deeper understanding and development in this area.

Based on the carefully constructed Tex-COIN dataset, we propose a strong baseline, **COINNet**, hinged on commonsense knowledge and non-parametric inference for cross-modal observation hypothesis inference. There are two proposed modules targeted for the characteristics of the COIN task: **(1) Knowledge-guided Cross-modal Alignment.** We adopt the supervised multi-task learning approach, which facilitates the model's understanding of commonsense knowledge pertaining to the video characters as described in the textual observation. This includes the alignment of heterogeneous visual information with the textual observation, such as character actions and scenes that are closely connected but temporally displaced from the textual observation. This perception of visual character actions lays the foundation for effective inference of the action flow. **(2) Graph-based Non-parametric Inference.** We develop a non-parametric action flow prediction module grounded in the principles of traditional graph theory. Specifically, during the training phase, we store the relationships between actions. In the testing step, we construct an action graph with the stored action relationships and apply the Dijkstra's algorithm [10] to locate the path connecting the actions described in the video and the textual observation. The experiments on the Tex-COIN dataset prove the efficacy of the COINNet model by significantly surpassing the state-of-the-arts.

Our contributions can be summarized as follows:

- We propose the Cross-modal Observation Hypothesis Inference task (**COIN**) by simulating human cognition. To the best of our knowledge, it is the early exploration of hypothesis inference in the multi-modal field.
- To facilitate research in this new domain, we contribute a large-scale dataset, Tex-COIN. This dataset consists of 39, 796

carefully annotated hypothesis inference examples. To further enrich the dataset and support heterogeneous alignment, we annotate commonsense knowledge for the key characters in the videos, including the character's appearance, clothing, actions, sentiment, etc.

- We introduce a strong baseline, COINNet, which facilitates heterogeneous alignment of visual information temporally displaced from the textual observation, and infers the action flow post the video event and preceding the textual event based on graph theory.
- Our experimental findings underscore the efficacy of the COINNet model. Through rigorous testing on the Tex-COIN dataset, COINNet significantly outperforms existing state-of-the-art models. In-depth analyses, including an ablation study and a case study, further validate the rationality of each module within COINNet, reinforcing its capacity to perform effectively in this challenging new task.

## 2 RELATED WORK

**Multi-modal Inference** With the development of the AI system [1, 9, 11, 28, 33], there are tremendous breakthroughs in the completion of perception tasks. However, further advancements are needed to complete multi-modal hypothesis inference tasks using AI systems. Lots of researchers are involved in this process [2, 6, 7, 18, 22, 31, 52]. In the area of the multi-modal hypothesis inference, some researchers introduce knowledge into auxiliary reasoning to complete [5, 8, 48, 56, 57]. [48] involves the external knowledge in the process of the visual question answer and achieves good results. In addition, lots of researchers focus on the fusion of cross-modal information [4, 17, 19, 32, 37]. [37] joints analysis of the input question and the input video with the convolutional graph to complete the video question answering task. Aligning the cross-modal information is widely adopted in the multi-modal inference domain [12, 23, 30, 54]. [12] aligns the cross-modal features with relevance affinity matrix. Compared with previous tasks, our COIN task has two different characteristics: **(1)** Our COIN task requires retrieving the target video from the video pool to explain the observation; **(2)** COIN needs to rigorously recover the complete action flow. They determine the distinctiveness of our Tex-COIN dataset, targeted for our task.

**Cross-modal Transformer.** Transformer-based models are becoming the mainstream algorithms of more and more cross-modal tasks [26, 27, 29, 50]. [44] introduces the transformer architecture

into the representation learning of the text and image. Many researchers [35, 36] bring the pre-trained model into different tasks (like text-video grounding) and achieve good performance. [55] decouples the spatio-temporal reasoning of the video effectively reducing the number of parameters inside the transformer. Similarly, [50]adopts the transformer model to complete the cross-modal spatial-temporal grounding task and achieves good performance. [55] introduces the parameters of the pre-trained image-text grounding model into its transformer structure and improves the model performance.

## 3  HYPOTHESIS INFERENCE DATASET

The **C**ross-modal **O**bservation hypothes**I**s i**N**ference task (**COIN**) aims to explain the textual observation by predicting an event chain (cause), which contains two sub-tasks: (A) retrieving the video from a video pool as the event chain starting point; (B) predicting the subsequent action flow after the retrieved video to complete the event chain. Towards two subtasks, there are two characteristics of our COIN task: Heterogeneous Alignment for Inference and Action Flow Inference. To promote the development of this field, we propose a large-scale dataset, **Tex-COIN**, which contains the commonsense knowledge annotation assisted to the cross-modal alignment training and the hypothesis inference example annotation. In this section, we will describe it in detail.

### 3.1  Dataset Overall

**Task Formulation.** Our Tex-COIN dataset is designed specifically for the characteristics of the COIN task. Thus, it is necessary to formulate the COIN task, before introducing our Tex-COIN. Given a textual observation (a language query) $O$ and a video pool $\mathcal{V} = \{\mathcal{V}_i\}_{i=1}^{N_\mathcal{V}}$ containing $N_\mathcal{V}$ videos, our COIN task aims to explain the textual observation $O$ by retrieving the target video from the video pool $\mathcal{V}$ and inferring the intermediate action flow $\mathcal{A} = \{\mathcal{A}_i\}_{i=1}^{N_\mathcal{A}}$ between the target video $\mathcal{V}_t$ and the textual observation $O$. With the AI system for our COIN task denoted as $\mathcal{M}$ with the parameter $\Theta$, the optimization function $\delta$ of the AI system $\mathcal{M}$ can be expressed as follows:

$$\delta(O, \mathcal{V}; \Theta) = \max_{\Theta} \xi(\mathcal{M}(O, \mathcal{V}; \Theta), \epsilon(O, \mathcal{V})). \tag{1}$$

In it, $\Theta$ is a learnable parameter. The AI system $\mathcal{M}(.)$ generates the model prediction and the function $\epsilon(.)$ outputs the ground truth. Then, the function $\xi(.)$ calculates the consistency of model prediction and the ground truth, which represents the loss calculation process described in Section 4.1.

**Dataset Construction Pipeline.** As shown in Figure 2, we summarize the dataset construction pipeline into three steps: **(A) Data Collection.** We collect the videos used for labeling from two sources, including lifestyle videos and TV show videos. Further, the annotators screen out the unqualified videos one by one. We add the detail to the appendix. **(B) Commonsense Knowledge Annotation.** The annotators label the commonsense knowledge (appearance, clothing, action, etc.) of the key characters in the collected videos. Then, the verifiers carefully check all the labels. **(C) COIN Task Annotation.** The labeling process for the COIN Task Annotation is similar to the Commonsense Knowledge labeling and consists of manual labeling and verification. In addition, we add additional annotations for

| Category | Subcategory |
|---|---|
| Clothing | Length of Lower-body Clothing, Type of Lower-body Clothing, Type of Upper-body Clothing, Sleeve Length, 3 Other Outfits, 9 Colors of Upper-body Clothing, 9 Colors of Lower-body Clothing |
| Appearance | Gender, Hair Length, Age |
| Sentiment | No Subcategories |
| Scene | No Subcategories |
| Action | Intransitive Verb, Transitive Verb, Object |

**Table 1:  Statistics of commonsense knowledge types for Tex-COIN.**

the text-video retrieval task to widen the dataset function, which is introduced in the appendix.

### 3.2  Dataset Construction

**Data Collection.** Before illustrating the annotation process, we introduce how to collect the data in our Tex-COIN. Specifically, The data in our Tex-COIN dataset comes from two sources: (A) We manually crop 10,000 video clips from 92 well-known TV shows, including Mr. Bean, Grey's Anatomy, etc. In addition, we carefully select 10,000 video clips from the previous datasets, including the HC-STVG dataset [51], the ava-actions dataset [21], and the TO-MAR dataset [34]. (B) 20,000 lifestyle videos are carefully collected from the Hwd dataset [49] and the vidor dataset [47]. Not all the collected videos are suitable for annotation in the next step, like the videos with extremely poor clarity, the videos with frequent transitions between story segments, the videos with long, meaningless static shots, etc. Thus, to construct the high-quality dataset, we carefully filter out the low-quality videos before annotation.

**Commonsense Knowledge Annotation.** With the filtered videos, we annotate the commonsense knowledge of the key characters in each video, including the characters' appearance, clothing, actions, sentiments, and located scenes. There are many subcategories in each type of commonsense knowledge, as shown in Table 1. To ensure high-quality labels, we divide the entire process into two parts, annotation and verification: **(A) Annotation.** Considering a large number of detailed labels required manual annotation, we organize a team of 20 annotators to complete the labeling. Among them, one annotator is responsible for the management, and 19 annotators are responsible for labeling. The entire labeling process takes 10 months to complete. **(B) Verification.** 4 verifiers are responsible for the verification of each annotated example, if all of them agree with the annotation, the example is saved. Otherwise, it is re-labeled and discarded.

**COIN Task Annotation.** Based on the filtered videos, 5 annotators are responsible for the labeling of the COIN task examples, which are undergraduate and graduate students from the top 50 universities in the QS World University Rankings. **(A) Annotation.** During the annotation process, the annotators first understand the

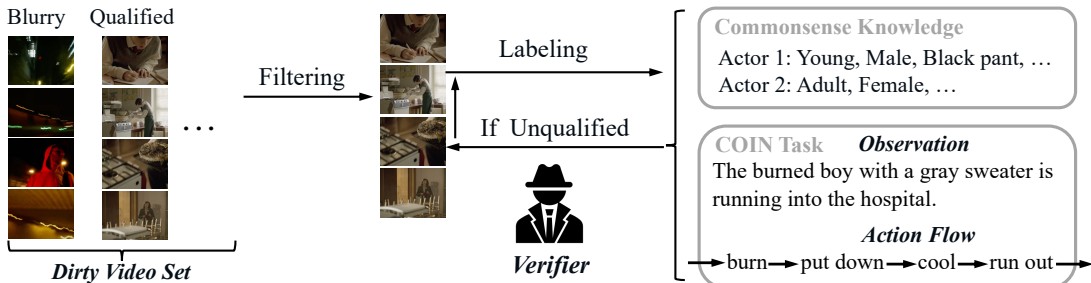

**Figure 2: A diagram of the dataset construction pipeline.**

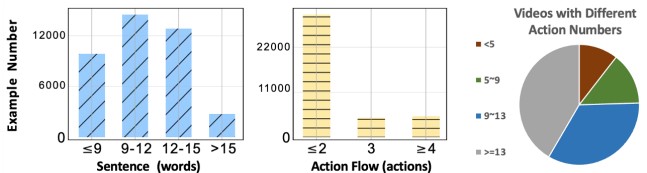

**Figure 3: The statistics of the Tex-COIN dataset.**

logic chains in the videos based on the visual information and the annotated commonsense knowledge. Then, the continuous clips are cut out from the videos and annotated with the natural language description of the observations $O$ by annotators referring to the follow-up video content and the commonsense knowledge. The subsequent action flow annotations $\mathcal{A}$ between the videos $\mathcal{V}$ and the textual observations $O$ are also recorded. **(B) Verification.** Same as the verification of commonsense knowledge, 2 verifiers with strong logical abilities are responsible for verifying the labeled examples. The labeled examples that have unanimous agreement are accepted. In addition, the language-based retrieval uniqueness needs to be guaranteed, which is added to the appendix.

## 3.3 Dataset In-depth Analysis

**Dataset characteristics** We are interested in the characteristics of the dataset. Therefore, we conduct a detailed analysis of the dataset, which is summarized as follows: **(A) Large-scale.** We propose a large-scale dataset for the COIN task, which consists of 39,796 hypothesis inference examples. We randomly split it into the non-overlapping train/val/test set (26,904/6,000/6,892). **(B) Diversity.** There are two types of data sources, including real daily-life data sources and TV show data sources. We manually filter out the high-quality data to guarantee that there are enough diverse people, actions, and scenes. At the same time, we also try our best to ensure that the logical chains in the videos are clear and diverse in the process of data selection.

**Dataset Statistics.** We conduct a statistical analysis of our Tex-COIN. The length statistics of the natural language queries (Figure 3) highlight the diverse amount of information conveyed through different language queries. In addition, from the statistics of the videos with different action numbers (Figure 3), there are significant variations in the video action number. It reflects that our dataset covers

videos with varying levels of content richness (action richness). We add more statistical results and analysis to the supplementary.

## 4 METHOD

Based on the novel task, **C**ross-modal **O**bservation hypothes**I**s i**N**ference (**COIN**), we introduce a carefully designed strong baseline, **COIN-Net**, whose diagram is shown in Figure 4. In this section, we will describe our COINNet in detail.

**Model Pipeline.** Our COINNet judges whether the given video $\mathcal{V}_i$ is part of the explanation of the textual observation $O$. Then, we choose the most suitable video with the highest probability from the video pool $\mathcal{V}$ and output the subsequent action flow as the explanation of the textual observation $O$. In detail, the model pipeline consists of three steps: **Step 1:** The **Knowledge-guided Cross-modal Alignment** module analyzes the input video $\mathcal{V}_i$ and the textual observation $O$, and predicts the commonsense knowledge (action, etc.) of the video character described by the textual observation $O$. **Step 2:** The **Graph-based Non-parametric Reasoning** module builds up the action knowledge graph according to the training examples and predicts the target action flow between the video action and the textual action with Dijkstra's algorithm [10]. **Step 3:** The **Reasoning Path Review** module analyzes the predicted action flow and the cross-modal feature output by the knowledge-guided cross-modal alignment module, to reason out the matching probability between the video $\mathcal{V}_i$ and the observation $O$. The best matching video is selected from the candidate video pool as the prediction.

## 4.1 Knowledge-guided Cross-modal Alignment

The alignment of the cross-modal information is fundamental to text-video retrieval [24, 40], which is an important subtask of our COIN. Thus, we propose the targeted knowledge-guided cross-modal alignment module. This module is trained in a supervised manner and could precisely predict the key commonsense knowledge (appearance, clothing, action, etc.) of the target video person described by the textual observation $O$.

Specifically, the knowledge-guided cross-modal alignment module is designed based on the transformer model [53]. The video feature $\mathbf{F}_v$ is extracted from the given video $\mathcal{V}_i$ and the language feature $\mathbf{F}_t$ is extracted from the textual observation $O$, with the visual encoder [13] and the language encoder [53]. Then, two types of query vectors are defined to analyze the multi-modal feature $\mathbf{F}_v$

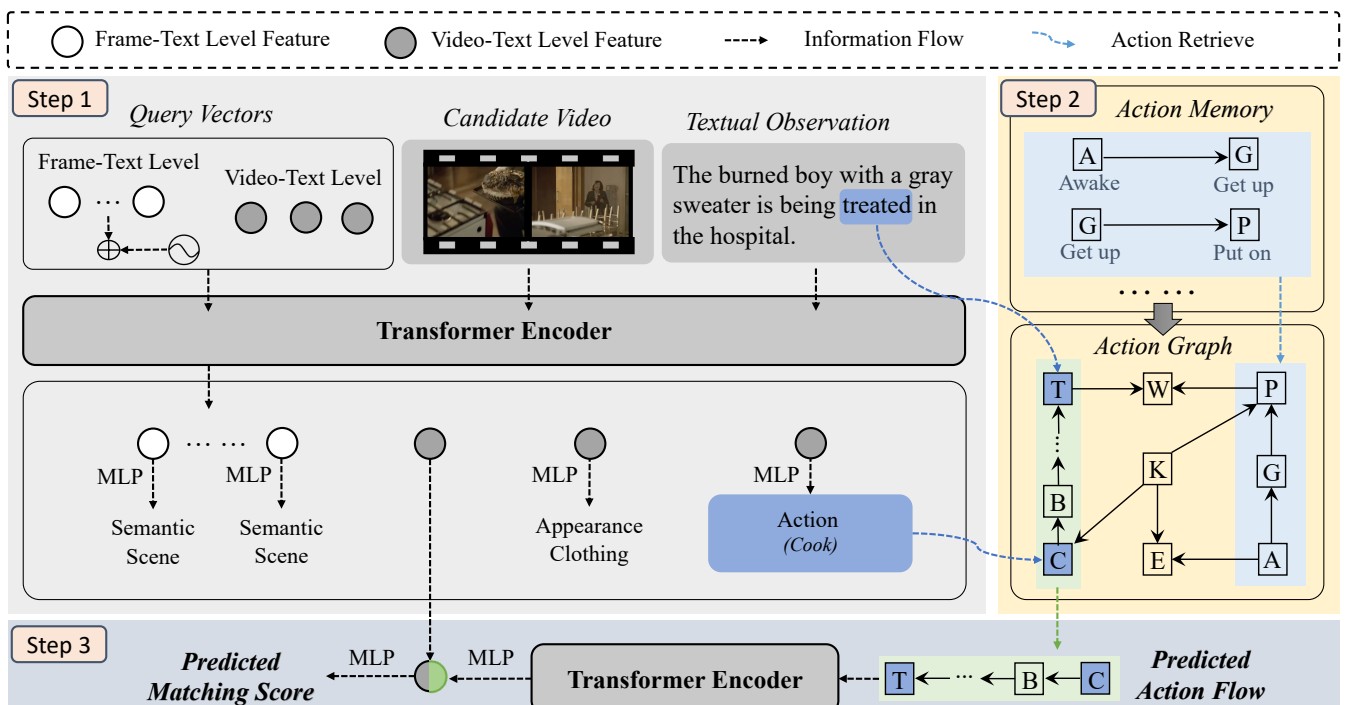

**Figure 4: The diagram of our proposed COINNet for the COIN task, which consists of three steps. Step 1: The Knowledge-guided Cross-modal Alignment module extracts the cross-modal feature and predicts the commonsense knowledge. Step 2: The Graph-based Non-parametric Reasoning model constructs the action graph based on action memory and predicts the action flow between the video action and the textual action. Step 3: The Inference Path Review module re-checks the predicted action flow and predicts the matching score between the video and the text observation.**

and $\mathbf{F}_t$ using the transformer encoder, which includes the frame-text level queries $\mathbf{Q}_f = \{\mathbf{q}_f^i\}_{i=1}^{N_f}$ and the video-text level queries $\mathbf{Q}_v = \{\mathbf{q}_v^i\}_{i=1}^{3}$. $N_f$ is the number of frames in the video $\mathcal{V}_i$. We inject the type embeddings to help the transformer encoder distinguish different types of query vectors. Then, the transformer encoder reasons out the corresponding features (frame-text level features $\mathbf{F}_f = \{\mathbf{f}_f^i\}_{i=1}^{N_f}$ and video-text level features $\mathbf{F}_v = \{\mathbf{f}_v^i\}_{i=1}^{3}$) relying on the corresponding query vectors ($\mathbf{Q}_f$ and $\mathbf{Q}_v$). After analyzing features output by the transformer encoder ($\mathbf{F}_f$ and $\mathbf{F}_v$), we predict all commonsense knowledge.

**(1) Sentiment and Scene Prediction.** Considering that the sentiment knowledge and the scene knowledge are changed over time they are predicted frame by frame with the frame-level features $\mathbf{F}_f$. We use the $i$-th frame as an example, and then the prediction process for the $i$-th frame with the corresponding frame feature $\mathbf{f}_f^i$ denoted as:

$$\mathbf{p}_{se}^i = \text{softmax}(MLP_{se}(\mathbf{f}_f^i)), \quad (2)$$

$$\mathbf{p}_{sc}^i = \text{softmax}(MLP_{sc}(\mathbf{f}_f^i)). \quad (3)$$

In them, $MLP_{se}$ and $MLP_{sc}$ are the **M**ulti**L**ayer **P**erceptron (**MLP**), which are applied to predict the sentiment and scene probabilities ($\mathbf{p}_{sc}^i$ and $\mathbf{p}_{se}^i$), respectively.

**(2) Appearance and Clothing Prediction.** Regarding the appearance and clothing knowledge of the target person in the video, they

are approximated as remaining constant throughout the video and are applied with the video-level query $\mathbf{f}_v^2$ to predict. In rare cases where there are special situations (such as costume changes), we require the model to predict the most relevant appearance and clothing knowledge for the textual observation description. We can represent the prediction process for the appearance and clothing knowledge as follows:

$$\mathbf{p}_{ap} = \text{softmax}(MLP_{ap}(\mathbf{f}_v^2)), \quad (4)$$

$$\mathbf{p}_{cl} = \text{softmax}(MLP_{cl}(\mathbf{f}_v^2)). \quad (5)$$

In the process, $MLP_{ap}$ and $MLP_{cl}$ are the MLP utilized to predict the appearance and clothing probabilities ($\mathbf{p}_{ap}$ and $\mathbf{p}_{cl}$), respectively.

**(3) Action Prediction.** When it comes to action knowledge, we also do not make frame-by-frame predictions as we do for sentiment knowledge and scene knowledge. It is due to the presence of multiple actions for the target person in each frame. Making frame-by-frame predictions would impose a heavy burden on our model and make it difficult to train effectively. As a result, the model directly predicts the target action most related to the textual observation $O$, instead of counting all video actions for each frame individually. We can represent the process of predicting the target video action using the video-level query feature $\mathbf{f}_v^1$ as follows:

$$\mathbf{p}_{ac} = \text{softmax}(MLP_{ac}(\mathbf{f}_v^1)). \quad (6)$$

In it, $MLP_{ac}$ is the MLP used for the target action prediction. $\mathbf{p}_{ac}$ represents the action type probability of the target action.

The model learns to make all commonsense knowledge predictions simultaneously. To achieve this, we use the cross-entropy loss function to calculate the losses for all commonsense knowledge, including $l_{se}$ (sentiment), $l_{sc}$ (scene), $l_{ac}$ (action), $l_{ap}$ (appearance), and $l_{cl}$ (clothing). To obtain the complete knowledge-guided loss $l_{total}$, we sum up all the individual losses mentioned above:

$$l_{total} = l_{ac} + l_{se} + l_{sc} + l_{ap} + l_{cl}. \tag{7}$$

## 4.2 Graph-based Non-parametric Inference

Due to the interdependent and interconnected actions in the action flow, the action flow inference subtask requires high correctness in learning the relationship between the actions and precisely predicting each action. Thus, we propose the targeted module, Graph-based Non-parametric Inference, which stores the action relationships to construct the action knowledge graph and finds the target action flow from the constructed graph with Dijkstra's algorithm.

Specifically, we represent the action set as $\mathcal{A} = \{\mathcal{A}_i\}_{i=1}^{N_{\mathcal{A}}}$. The module diagram is shown in Figure 4. From it, we can find that there are two steps in the graph-based non-parametric inference module: **(1) Training Step.** During the training process, we *split* the action flow labeled for each example into multiple single-step relational maps and store them in the action memory. No trainable parameters are involved in this process and we don't need to further train the action memory module. We take the action flow $\mathcal{A}_1 \rightarrow \mathcal{A}_2... \rightarrow \mathcal{A}_r$ as an example to present this process, which is represented as:

$$(\mathcal{A}_1 \rightarrow \mathcal{A}_2), ..., (\mathcal{A}_{r-1} \rightarrow \mathcal{A}_r) = split(\mathcal{A}_1... \rightarrow \mathcal{A}_r). \tag{8}$$

**(2) Testing Step.** At the testing step, the action knowledge graph $\mathcal{G}(\mathcal{A}, \mathcal{U})$ is constructed by *connecting* all the single-step relational maps $(\mathcal{A}_i \rightarrow \mathcal{A}_{i+1}), ..., (\mathcal{A}_{i+n-1} \rightarrow \mathcal{A}_{i+n})$ in the memory generated in the step (1). Here, we define $\mathcal{U}$ as the edges between the actions in the action knowledge graph. With the constructed action knowledge graph, we find the target path as the predicted action flow during the testing process. Firstly, the starting action knowledge node $\mathcal{A}_s$ and the ending action knowledge node $\mathcal{A}_e$ of the target path (target action flow) are found in the constructed graph $\mathcal{G}(\mathcal{A}, \mathcal{U})$. We find the starting node $\mathcal{A}_s$ by matching the action predicted by the knowledge-guided cross-modal alignment module (described in Section 4.1) and each action knowledge node in the graph $\mathcal{G}(\mathcal{A}, \mathcal{U})$. Similarly, the ending node $\mathcal{A}_e$ is found by matching each action knowledge node in the graph $\mathcal{G}(\mathcal{A}, \mathcal{U})$ and the action of the textual observation $O$. The textual action is detected with the widely applied tool, StanfordNLP [41]. Secondly, we use the Dijkstra's algorithm to find the connected path between the starting node $\mathcal{A}_s$ and the ending node $\mathcal{A}_e$. We formalize the path-finding process as:

$$\mathcal{A}_s... \rightarrow \mathcal{A}_e = Dijkstra(\mathcal{A}_s, \mathcal{A}_e, \mathcal{G}(\mathcal{A}, \mathcal{U})). \tag{9}$$

After finding the connected path, the found action flow (connected path) is preserved and needs to be further verified in the next reasoning path review module.

## 4.3 Inference Path Review

As the key module in our COINNet model, this module reasons out the matching score between the given video $\mathcal{V}$ and the textual observation $O$, according to the cross-modal feature $\mathbf{f}_v^0$ and the predicted action flow $\mathcal{A}_s... \rightarrow \mathcal{A}_e$.

Specifically, we first encode the found action flow $\mathcal{A}_s... \rightarrow \mathcal{A}_e$, with a new transformer encoder different from the one in the Knowledge-guided Cross-modal Alignment module. Then, we get the feature representation $\mathbf{f}_{\mathcal{A}}$ of the action flow. We predict the cross-modal matching score with the video-text feature $\mathbf{f}_v^0$ (containing the information from the given video $\mathcal{V}$ and the textual observation $O$) and the action flow feature $\mathbf{f}_{\mathcal{A}}$:

$$\mathbf{p}_{match} = \text{softmax}(MLP_{match}([\mathbf{f}_v^0, \mathbf{f}_{\mathcal{A}}])), \tag{10}$$

where $\mathbf{p}_{match}$ is the probability vector of the cross-modal matching or not. [.] represents the feature concatenation. In addition, $MLP_{match}(.)$ represents the multi-layer perception and softmax(.) is the activation function. If there are multiple found action flows in step 2 of our COINNet model, we select the one corresponding to a higher predicted matching score as the final result.

## 5 EXPERIMENT

In this section, we will evaluate our proposed baseline COINNet on our Tex-COIN dataset.

**Baselines.** Previous methods designed for other tasks cannot be applied to solve our COIN task, directly. Thus, we extend several state-of-the-arts to compare. In detail, for a comprehensive comparison, we introduce the following methods of similar format tasks: (1) text-video retrieval methods, including **DMAE** [25], **MASCOT** [15], **CRET** [24], **Clip4Clip** [40] (2) action flow prediction methods, including **CycleC** [16], **FUTR** [20], and **VLMAH** [42]. During the performance comparison, we pairwise fuse them to form 10 baselines: **DMAE_C** (DMAE + CycleC), **DMAE_F** (DMAE+FUTR), **MASCOT_C** (MASCOT + CycleC), **MASCOT_F** (MASCOT+FUTR), **MASCOT_V** (MASCOT+VLMAH), **CRET_C** (CRET + CycleC), **Clip4Clip_C** (Clip4Clip + CycleC), **CRET_F** (CRET + FUTR), **Clip4Clip_F** (Clip4Clip + FUTR), **Clip4Clip_V** (Clip4Clip + VLMAH).

**Implement Detail.** We implement our COINNet on a Linux server with Pytorch 1.4 and 8 Tesla V100 with 32GB memory. During the training process, we set the training rate as $1e-5$. The batch size is set to 8 and the optimizer is AdamW. In addition, we also employ some data augmentations, such as random horizontal flipping, random cropping, and so on. We adopt the same pretraining parameters for the COINNet as the Clip4Clip baseline [40].

**Evaluation Metrics.** Following the widely used evaluation protocol of the text-video retrieval task [24, 39], **Rank N** (R@N, N=1,5,10) and median rank (**MdR**) are applied to evaluate the model performance for the explanation video retrieval. We adopt **MoC** used by the future action prediction tasks [16] to evaluate the action flow predicted results In addition, we also apply accuracy (**ACC**) to evaluate the model performance for the action flow prediction, which is added in the appendix.

| Methods | R@1 | R@5 | R@10 | MdR | MoC |
|---|---|---|---|---|---|
| **DMAE_C** | 3.4 | 15.1 | 21.1 | 32 | 24.1 |
| **DMAE_F** | 3.1 | 15.2 | 22.1 | 31 | 25.8 |
| **CRET_C** | 3.5 | 13.1 | 20.1 | 33 | 25.4 |
| **CRET_F** | 3.3 | 13.9 | 21.5 | 32 | 27.6 |
| **MASCOT_C** | 4.7 | 16.5 | 28.6 | 28 | 26.9 |
| **MASCOT_V** | 5.0 | 16.2 | 29.8 | 27 | 27.0 |
| **MASCOT_F** | 5.1 | 17.2 | 30.1 | 27 | 27.3 |
| **Clip4Clip_C** | 5.4 | 20.5 | 33.1 | 20 | 26.1 |
| **Clip4Clip_V** | 5.8 | 20.6 | 34.7 | 21 | 27.8 |
| **Clip4Clip_F** | 5.9 | 21.1 | 35.0 | 19 | 28.0 |
| **COINNet** | **7.3** | **25.3** | **39.9** | **16** | **31.9** |

**Table 2: Comparing our strong baseline, COINNet, with other baselines on the Tex-COIN dataset. We color the best results.**

| Methods | R@1 | R@5 | R@10 | MdR | MoC |
|---|---|---|---|---|---|
| **Base Model** | 5.7 | 21.5 | 34.9 | 20 | 27.9 |
| $+\delta_{Align}$ | 6.3 | 23.1 | 36.8 | 17 | 30.1 |
| $+\delta_{Reason}$ | 6.9 | 23.5 | 37.1 | 17 | 31.8 |
| **COINNet** | **7.3** | **25.3** | **39.9** | **16** | **31.9** |

**Table 3: Ablation study on the Tex-COIN dataset. $\delta_{Align}$ represents the Knowledge-guided Cross-modal Alignment module. $\delta_{Reason}$ represents the Graph-based Non-parametric Inference module. We color the best results.**

## 5.1 Performance Comparison

We are interested in the COINNet performance compared with the state-of-the-arts. The comparison results are shown in Table 2. From it, we have the following findings:

- It can be found that our COINNet model performs better than other baselines. We attribute the performance improvement to the cross-modal precise alignment with the knowledge-guidance, and the action-relation learning and inference ability of the Graph-based Non-parametric module.
- The baselines, like **DMAE_C** and **CRET_C**, are extended from other baselines and lack the domain knowledge, which limits the performance of these baselines.

In addition, we fully compare our COINNet with the generative large language model in the appendix, which proves the same conclusion.

## 5.2 Ablation Study

We are interested in the effect of each COINNet module. Thus, we design the ablation study by surgically removing these modules,

| Methods | R@1 | R@5 | R@10 | MdR | MoC |
|---|---|---|---|---|---|
| **Base Model** | 5.7 | 21.5 | 34.9 | 20 | 27.9 |
| $+\delta_{appreance}$ | 6.0 | 22.9 | 35.1 | 18 | 28.7 |
| $+\delta_{clothing}$ | 5.9 | 22.3 | 35.4 | 19 | 28.9 |
| $+\delta_{action}$ | 6.2 | 22.8 | 35.8 | 18 | 29.4 |
| $+\delta_{sentiment}$ | 5.8 | 21.9 | 36.3 | 19 | 29.8 |
| $+\delta_{scene}$ | 5.8 | 21.7 | 35.6 | 20 | 29.5 |
| $+\delta_{Align}$ | 6.3 | 23.1 | 36.8 | 17 | 30.1 |

**Table 4: Ablation study on the Tex-COIN dataset about the commonsense knowledge. It is worth noting that each commonsense knowledge is individually added to the base model and tested. $+\delta_{Align}$ represents adding the guidance of all the commonsense knowledge together.**

including the knowledge-guided Cross-modal Alignment and the Graph-based Non-parametric Inference. After removing the Graph-based Non-parametric Inference module, the model cannot perform the action flow prediction. Therefore, we choose to replace this non-parametric module with the FUTR model, instead of completely removing it.

The experiment results are shown in Table 3. From it, we have the following findings:

- After adding any module proposed by us, the ablation base model performs better. It proves the effectiveness of each module.
- When adding the Knowledge-guided Cross-modal Alignment module, the Graph-based Non-parametric Inference module performs better. This demonstrates that guided by knowledge, our COINNet model has a stronger cross-modal alignment ability. It leads to a more accurate prediction of key actions in videos related to textual observation, which is essential for the Graph-based Non-parametric Inference module.

In addition, we add the guidance of each commonsense knowledge independently (including appearance, clothing, action, sentiment, and scene), to evaluate the effectiveness of each one. The experiment results are shown in Table 4. From it, we can observe that the model performs better after adding each of the commonsense knowledge guidance. It proves the reasonable design of the Knowledge-guided Cross-modal Alignment. Finally, we analyze the COINNet performance solely relying on the character's physical description to retrieve the target video and add the experiment results to the appendix.

## 5.3 In-depth Analysis

**Comparison with baselines on different training data volumes.** In order to assess the performance of the COINNet model trained on varying amounts of data, we conduct experiments by randomly selecting 25%, 50%, 75%, and 100% of the training data from movie scenes. The comparison results between COINNet and the Clip4Clip_F baseline are presented in Figure 5(a). Notably, benefiting from our

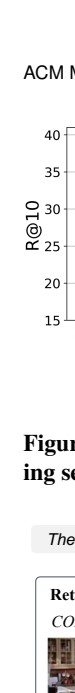

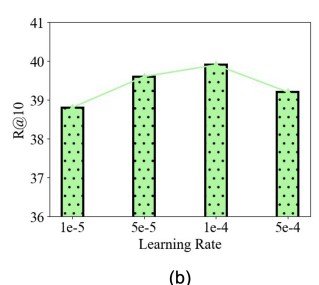

**Figure 5: Baseline comparison on different proportions of training set.**

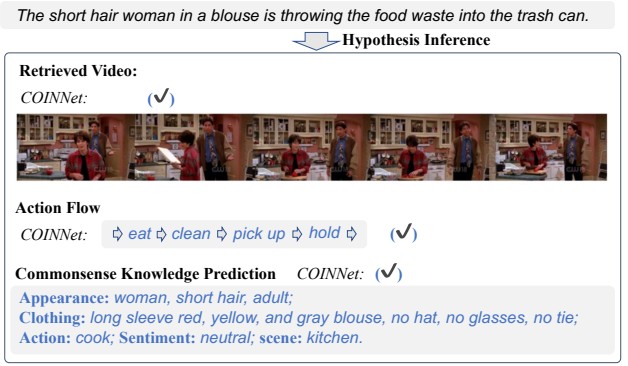

**Figure 6: An example of our COINNet baseline predictions.**

proposed modules, the accuracy of COINNet remains higher than that of the Clip4Clip_F baseline, even with a smaller training data volume.

**Hyperparameter Analysis** We are interested in the effectiveness of the training hyperparameters. Thus, we select the key hyperparameter, the learning rate, for further analysis. The experiment results are shown in Figure 5(b). Notably, the COINNet model achieves the highest accuracy with a learning rate of 1e-4, which is adopted by us. In addition, our method, COINNet, has low sensitivity to learning rate, which reflects its robustness.

**Case Study** To further demonstrate the effectiveness of our COIN-Net, we randomly select an example from the Tex-COIN dataset and visualize the prediction of our COINNet baseline. The experiment results are shown in Figure 6. In the shown case, the complete event process is: The woman cooked in the kitchen. → She ate the food. → She cleaned the kitchen waste. → She picked up, held, and threw the food waste. It can be observed that our strong baseline, COINNet, precisely predicts the target video and the action flow, and reasons out all commonsense knowledge correctly, which proves the reasonable design of our COINNet. More examples are shown in the appendix.

### 5.4 Text-video Retrieval Task

We add the text-video retrieval annotations for our Tex-COIN dataset to extend the dataset function, and experiment with the retrieval baselines on the Tex-COIN dataset. In this section, we will describe them in detail.

| Methods | R@1 | R@5 | R@10 | MdR |
|---|---|---|---|---|
| MASCOT | 4.9 | 19.0 | 31.0 | 24 |
| Clip4Clip | 5.2 | 19.5 | 32.7 | 21 |
| Base Model | 6.0 | 19.7 | 32.1 | 21 |
| COINNet (Base Model + $\delta_{Align}$) | 6.5 | 20.5 | 34.0 | 18 |

**Table 5: Comparing our model, COINNet, with other baselines on the Tex-COIN dataset for the text-video retrieval task. $\delta_{Align}$ represents the Knowledge-guided Cross-modal Alignment design.**

**Dataset Annotation.** Following the COIN task annotation, we provide the annotations for the text-video retrieval task on the Tex-COIN dataset. Specifically, 4 trained annotators are responsible for the annotation. The process consists of two steps: (1) The annotators view all videos in the Tex-COIN dataset. Then, the key clips of the videos are cropped out and labeled with the language description. (2) 2 verifiers are responsible for conducting detailed checks on the annotations. If they do not agree with the label, it is re-labeled or discarded. After annotation, there are 9,983 examples in our Tex-COIN dataset for the text-video retrieval task, which are split into train/val/test (9,033/200/750). More details about the annotations are shown in the supplementary.

**Performance Comparison.** We compare our COINNet model with the state-of-the-art on Tex-COIN dataset with the text-video retrieval annotations. In addition, we conduct the ablation study for our COINNet. The experiment results are shown in Table 5. From it, it can be found that our COINNet model performs best. We attribute the improvement to the effectiveness of the Knowledge-guided Cross-modal Alignment design.

## 6 CONCLUSION

In this paper, we propose a new task (COIN), the task-toward dataset (Tex-COIN), and the targeted strong baseline (COINNet), to promote the improvement of hypothesis inference in the multi-modal field. Experiments on the Tex-COIN prove the effectiveness of our COINNet model. We believe that our work can assist in building the human-like reasoning AI system and help to improve its performance in hypothesis-inference applications. For instance, in intelligent security, given the description of the witness, AI system with the hypothesis inference capability can search for visual evidence from surveillance and infer the potential criminal process. Although some progress has been made, there are still several limitations left for future work: (1) The model accuracy is far from reaching its upper limit. (2) Following the previous datasets [38, 45], we collect data from the real-life and movie domains. More data from other domains can further enrich our dataset. In addition, we believe that our work can assist in building the human-like reasoning AI system and help to improve its performance in hypothesis-inference applications. For instance, in intelligent security, given the description of the witness, the AI system with the hypothesis inference capability can search for visual evidence from surveillance and infer the potential criminal process.

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
