# OpenReview forum: "Cross-modal Observation Hypothesis Inference"
_acmmm.org/ACMMM/2024/Conference — MM2024 Oral_

### Official Review · Reviewer_gaVd · 2024-05-23

**Rating:** 5
**Confidence:** 3

**Summary:**

The paper introduces a novel task called Cross-modal Observation Hypothesis Inference (COIN), aiming to simulate human cognitive processes in multi-modal AI systems. The authors present a large-scale dataset, Tex-COIN, and a strong baseline model, COINNet, which integrates commonsense knowledge and non-parametric inference for cross-modal hypothesis inference. The method is validated on Tex-COIN. SOTA performance is achieved.

**Strengths:**

1.  The paper introduces a new and challenging task focusing on cross-modal observation hypothesis inference through the alignment of textual observations with visual content.

2. Experiments validate the superior performance of COINNet over existing state-of-the-art models.

3. The paper is well organized and easy to follow. The idea and the method are well presented.

**Limitations:**

1.   The baseline comparisons may not be comprehensive enough to demonstrate the full efficacy of the proposed model.  The experiments on the Tex-COIN dataset focus primarily on comparing COINNet with state-of-the-art models **in a limited set of scenarios**. Expanding the scope to include more diverse and challenging benchmarks would strengthen the evaluation. And the detailed structure and parameters of other models should be contained for a fair comparison.

2. COINNet seems to primarily consider temporal relations among objects and descriptive questions within the video. However, deep inference capabilities, including causal reasoning, are crucial for comprehensive video understanding. Does the current task setup truly challenge these advanced inference abilities?


3. Could **the heavy reliance on manual annotations introduce biases and scalability issues**? The Tex-COIN dataset contains 39,796 meticulously annotated hypothesis inference examples by annotators with strong logical abilities. Might this reliance on manual annotation processes lead to inconsistencies and pose challenges for scaling the dataset effectively?


4. What about the complexity or time cost? Providing an analysis of the computational complexity and the time required for training and inference, compared to other methods, would offer valuable insights into the practical applicability of COINNet.

**Suitability:**

3

---

### Official Review · Reviewer_7zeQ · 2024-05-24

**Rating:** 4
**Confidence:** 3

**Summary:**

The paper introduces a new task called Cross-modal Observation Hypothesis Inference (COIN), which involves creating a system that can recall events from a visual memory to explain partially observed textual descriptions. The proposed system, COINNet, uses a novel dataset, Tex-COIN, containing text-video examples and integrates commonsense knowledge to enhance inference accuracy. Extensive experiments validate the effectiveness of COINNet, which significantly outperforms existing models.

**Strengths:**

1. The COIN task is novel and taps into an underexplored area of multi-modal AI, merging hypothesis generation with visual and textual analysis.
2. The development of the Tex-COIN dataset, which includes diverse examples and commonsense annotations, provides a solid foundation for training and evaluating the model.
3. COINNet integrates advanced techniques like transformer-based multi-task learning and graph-based inference, demonstrating superior performance over state-of-the-art models in comprehensive tests.

**Limitations:**

1. While the paper introduces a novel dataset and model, the generalizability of COINNet across different datasets and real-world scenarios remains unclear. The specialized nature of the Tex-COIN dataset might not represent the variety of contexts and environments where such a system would be deployed.
2. The development and training of COINNet potentially inherit biases present in the dataset. The paper does not address how it handles potential biases or the impact these biases could have on the model's performance.
3. A few minor errors. For example, with lines 426 and 677, the text content exceeds the limit. Please check the full text.

**Suitability:**

3

---

### Official Review · Reviewer_LVZW · 2024-05-28

**Rating:** 5
**Confidence:** 2

**Summary:**

This study proposes a cross-modal observation hypothesIs inference task named COIN, aiming to  recall the inception of an event from memory and infer a cohesive event flow that furnishes a robust explanation for an incomplete observation.
Based on the task, the study constructs a Tex-COIN dataset and introduces COINNet as baseline.

**Strengths:**

1. The cross-modal observation hypothesIs inference task is interesting and challenging. The hypothesIs inference capacity is valuable for modern AI. Heterogeneous alignment for inference and action flow inference make the task challenging.

2. The Tex-COIN dataset is well-contructed. The diagram of the dataset construction pipeline and the  statistics of dataset are clear.

3. The baseline for the COIN task is given named COINNet.

**Limitations:**

1. The evaluation criteria for this task are worth discussing.

Since it is motivated as a inference task, the evaluation of the inference results should be subjective. However, the widely used evaluation protocol of the text-video retrieval task and accuracy of prediction task  can only  reflect the performance of  model output matching with the dataset.

The potential issue is, if the model I propose can have good retrieval and predictive performance, then will the reasoning I output conform to human objective cognition, can it explain observations well, and even be logically coherent?

**Suitability:**

3

---

### Meta-Review · Area_Chair_P6ML · 2024-06-28

**Recommendation:** Accept (Oral)
**Confidence:** 5

**Metareview:**

All reviewers have expressed positive attitudes towards this paper, so we have decided to accept this paper.